# Cell-Culture Adaptation of H3N2 Influenza Virus Impacts Acid Stability and Reduces Airborne Transmission in Ferret Model

**DOI:** 10.3390/v13050719

**Published:** 2021-04-21

**Authors:** Valerie Le Sage, Karen A. Kormuth, Eric Nturibi, Juhye M. Lee, Sheila A. Frizzell, Michael M. Myerburg, Jesse D. Bloom, Seema S. Lakdawala

**Affiliations:** 1Department of Microbiology and Molecular Genetics, University of Pittsburgh School of Medicine, Pittsburg, PA 15219, USA; valerie.lesage@pitt.edu (V.L.S.); kkormuth@bethanywv.edu (K.A.K.); nturibi.eric@medstudent.pitt.edu (E.N.); 2Division of Basic Sciences and Computational Biology Program, Fred Hutchinson Cancer Research Center, Seattle, WA 98109, USA; juhyelee@uw.edu (J.M.L.); jbloom@fredhutch.org (J.D.B.); 3Department of Genome Sciences, University of Washington, Seattle, WA 98195, USA; 4Department of Medicine, Division of Pulmonary, Allergy, and Critical Care Medicine, University of Pittsburgh School of Medicine, Pittsburgh, PA 15219, USA; samfrizzell@icloud.com (S.A.F.); myerburgm@upmc.edu (M.M.M.); 5Howard Hughes Medical Institute, Seattle, WA 98103, USA; 6Center for Vaccine Research, University of Pittsburgh School of Medicine, Pittsburgh, PA 15219, USA

**Keywords:** influenza virus, transmission, pH, hemagglutinin

## Abstract

Airborne transmission of seasonal and pandemic influenza viruses is the reason for their epidemiological success and public health burden in humans. Efficient airborne transmission of the H1N1 influenza virus relies on the receptor specificity and pH of fusion of the surface glycoprotein hemagglutinin (HA). In this study, we examined the role of HA pH of fusion on transmissibility of a cell-culture-adapted H3N2 virus. Mutations in the HA head at positions 78 and 212 of A/Perth/16/2009 (H3N2), which were selected after cell culture adaptation, decreased the acid stability of the virus from pH 5.5 (WT) to pH 5.8 (mutant). In addition, the mutant H3N2 virus replicated to higher titers in cell culture but had reduced airborne transmission in the ferret model. These data demonstrate that, like H1N1 HA, the pH of fusion for H3N2 HA is a determinant of efficient airborne transmission. Surprisingly, noncoding regions of the NA segment can impact the pH of fusion of mutant viruses. Taken together, our data confirm that HA acid stability is an important characteristic of epidemiologically successful human influenza viruses and is influenced by HA/NA balance.

## 1. Introduction

Influenza A viruses cause acute respiratory disease in mammals and birds, whereas aquatic avian species and bats act as zoonotic reservoirs. Host tropism is determined by a combination of viral proteins and host factors which allow for efficient replication and transmission within a given species. Influenza virus particles have two major surface glycoproteins, hemagglutinin (HA) and neuraminidase (NA). The trimeric HA protein binds sialic acids on cell-membrane-bound proteins and is a major determinant of host tropism. Avian HA proteins preferentially bind to an α2,3-linked sialic acid, whereas human viruses have an α2,6-linked sialic acid preference [1,2,3]. HA is functionally balanced by the tetrameric receptor-destroying NA. At later stages of infection, NA plays a major role in removing sialic acids from host-cell receptors, as well as from newly synthesized HA and NA on nascent virions, which are sialylated during transport through the endoplasmic reticulum and Golgi apparatus [4,5]. Sialic acid removal by NA prevents virion aggregation and promotes spread to new target cells [5]. The functional balance between HA and NA is important to the maintenance of viral fitness, as NA needs to be active enough to disaggregate virions upon release but not so active that HA receptor attachment is decreased [6].

Proteolytic cleavage of the HA0 precursor by host proteases produces HA1 and HA2 subunits to reveal the fusion peptide, which is necessary for membrane fusion [7,8]. Once bound to sialic acids, influenza viruses are internalized via receptor-mediated endocytosis and HA mediates escape from early endosomes in a pH-dependent manner. At a fixed pH, HA undergoes an irreversible conformational change, which facilitates fusion of the viral and cellular membranes and the consequent release viral genomes into the host cytoplasm [9]. The pH stability of viral HA varies by host origin. The HA protein from circulating human influenza viruses undergoes a confirmation change at pH 5.0–5.4 [10,11,12]. In contrast, avian and swine influenza viruses are less acid stable, and their HA proteins undergo a conformational change at a pH > 5.5 [12,13,14,15,16,17,18,19]. Thus, HA pH of fusion is an important factor in host adaptation and transmissibility [12,20,21,22].

The importance of HA pH of fusion in the cell adaptation and transmission of H3N2 viruses is unclear. In this study, we demonstrated that cell adaptation of a seasonal 2009 H3N2 virus (A/Perth/16/2009) resulted in HA mutations (G78D and T212I) [23] that decreased HA acid stability. The adapted H3N2 virus with these two HA mutations (herein referred to as the ‘rPerth mutant’) exhibits enhanced replication in MDCK cells as compared to the wild-type virus (herein referred to as ‘rPerth WT’). However, no growth difference was observed in more physiologically relevant adenocarcinoma human alveolar basal epithelial (A549) and human bronchial epithelial (HBE) cells. In ferrets, the rPerth mutant replicated similarly to rPerth WT, but displayed reduced efficiency of transmission via respiratory droplets between ferrets. Our observations suggest that, similarly to H1N1 viruses, H3N2 viruses with a higher pH of fusion have decreased airborne transmission, confirming the importance of this phenotype for transmission fitness.

## 2. Material and Methods

### 2.1. Cells and Viruses

MDCK epithelial cells (ATCC, CCL-34) were maintained in Eagle’s minimal essential medium (MEM) supplemented with 10% fetal bovine serum (FBS), penicillin/streptomycin, and L-glutamine. A549 and 293T cells (ATCC, CCL-185 and CRL-11268) were maintained in Dulbecco’s MEM (DMEM) supplemented with 10% FBS, penicillin/streptomycin, and L-glutamine. Primary HBE cells derived from human lung tissue were differentiated and cultured at an air–liquid interface using a protocol approved by the relevant institutional review board [24]. The H3N2 viruses A/Panama/2007/2009, A/Perth/16/2009, A/Wyoming/3/2003, and A/Wisconsin/67/2005 were a generous gift from Dr Zhiping Ye (Center for Biologics Evaluation and Research, FDA, Silver Spring, MD, USA). A/Minnesota/11/2010 H3N2v and sw/OK/011506/2007 H3N2 viruses were a kind gift from Dr Kanta Subbarao (Doherty Institute, Melbourne, Australia). Influenza virus titers were determined by TCID_50_ assay on MDCK cells according to the method of Reed and Muench [25].

### 2.2. Plasmid-Based Reverse Genetics

The viruses used in this study were generated using bidirectional reverse-genetics plasmids based on pHW2000 [26]. The plasmids were cloned from cDNA reverse transcribed from the A/Perth/16/2009 (H3N2) virus stock described in the subsection immediately above, and were named pHW-Perth09-PB2-SL, pHW-Perth09-PB1-SL, pHW-Perth09-PA-SL, pHW-Perth09-HA-SL, pHW-Perth09-NP, pHW-Perth09-NA, pHW-Perth09-M, and pHW-Perth09-NS1. The noncoding regions of the pHW-Perth09-HA-SL and pHW-Perth09-NA-SL exactly match the noncoding regions of the comparable vRNAs deposited in Genbank by the WHO Collaborating Centre for influenza research in Australia (Genbank GQ293081 and GQ293082), except for a single A nucleotide inserted after the 21st nucleotide in the noncoding region at the 5′ end of the NA vRNA. We also used the variant HA and NA plasmids described in Reference [23] as pHW-Perth09-HA-G78D-T212I and pHW-Perth09-NA. These variant plasmids differ in two ways from the initial set of eight: they have noncoding regions from the lab-adapted X-31 strain, and the HA has two amino-acid mutations that were selected after passaging in cell culture (see Reference [23] for details). The virus termed ‘rPerth WT’ was generated from the pHW-Perth09-*-SL series of plasmids, whereas the virus termed ‘rPerth mutant’ used the pHW-Perth09-HA-G78D-T212I and pHW-Perth09-NA plasmids and the other six genes from the pHW-Perth09-*-SL series. The sequences of all plasmids are provided on FigShare at https://figshare.com/projects/Cell_Culture_Adaptation_of_H3N2_Influenza_Virus_Impacts_Acid_Stability_and_Reduces_Airborne_Transmission_in_Ferrets/112005(accessed on 21 April 2021). 

Recombinant viruses were generated using the eight reverse-genetics plasmids transfected into 293T cells, using TransIT-LT1 transfection reagent (Mirus Bio LLC, Madison, WI, USA) in accordance with the manufacturer’s protocol. After 24 and 48 h, 293T cell supernatants were used to infect MDCK cells to rescue a CP1 stock of virus.

### 2.3. Replication Kinetics

A549 and MDCK cells were infected with the indicated virus at a multiplicity of infection (MOI) of 0.01 or 1.0 (calculated on the basis of the TCID_50_) in MEM containing 2% L-glutamine and supplemented with 1 μg/mL L-(tosylamido-2-phenyl) ethyl chloromethyl ketone (TPCK)-treated trypsin (Worthington Biochemical, Lakewood, NJ, USA). Virus was allowed to adsorb for 1 h before the inoculum was removed and replaced with fresh medium. Supernatants were collected at the indicated time points and titered by TCID_50_.

Three different HBE patient cultures were used: HBE0176, HBE0256, and HBE259. The apical surface of the HBE cells was washed in phosphate-buffered saline (PBS), and 10^3^ TCID_50_ of virus was added per 100 μL of HBE growth medium. After a 1 h incubation at room temperature, the inoculum was removed and the apical surface was washed three times with PBS. At the indicated time points, 150 μL of HBE medium was added to the apical surface for 10 min to capture released virus particles. The experiment was performed in triplicate in three different patient cell lines.

### 2.4. Animal Ethics Statement

Ferret experiments were conducted in a BSL2 facility at the University of Pittsburgh in compliance with the guidelines of the Institutional Animal Care and Use Committee (approved protocols #16077170 and #19075697). Animals were sedated with isoflurane following approved methods for all nasal washes and survival blood draws. Ketamine and xylazine were used for sedation for all terminal procedures, followed by cardiac administration of euthanasia solution. Approved University of Pittsburgh Division of Laboratory Animal Resources (DLAR) staff administered euthanasia at time of sacrifice. 

### 2.5. Ferret Screening

Five to six month old male ferrets were purchased from Triple F Farms (Sayre, PA, USA). All ferrets were screened by hemagglutinin inhibition assay for antibodies against circulating influenza A and B viruses, as described in Reference [27], using the following antigens obtained through the International Reagent Resource, Influenza Division, WHO Collaborating Center for Surveillance, Epidemiology and Control of Influenza, Centers for Disease Control and Prevention, Atlanta, GA, USA: 2018–2019 WHO Antigen, Influenza A (H3) Control Antigen (A/Singapore/INFIMH-16-0019/2016), BPL-Inactivated, FR-1606; 2014–2015 WHO Antigen, Influenza A (H1N1)pdm09 Control Antigen (A/California/07/2009 NYMC X-179A), BPL-Inactivated, FR-1184; 2018–2019 WHO Antigen, Influenza B Control Antigen, Victoria Lineage (B/Colorado/06/2017), BPL-Inactivated, FR-1607; 2015–2016 WHO Antigen, Influenza B Control Antigen, Yamagata Lineage (B/Phuket/3073/2013), BPL-Inactivated, FR-1403.

### 2.6. Transmission Studies

Our transmission caging setup was a modified Allentown ferret and rabbit bioisolator cage similar to those used in our previous studies, described in References [27,28,29]. For each study, three ferrets were anesthetized with isoflurane and inoculated intranasally with 10^6^ TCID_50_/500 μL of A/Perth/16/2009 WT or mutant virus to act as donor animals. Twenty-four hours later, a recipient ferret was placed into the cage but separated from the donor animal by two staggered perforated metal plates welded together one inch apart. Recipients were exposed for 14 days. Nasal washes were collected from each donor and recipient every other day for 14 days. To prevent accidental contact or fomite transmission by investigators, the recipient ferrets were handled first and extensive cleaning of gloves, chambers, biosafety cabinet, and temperature monitoring wand was performed between each recipient and donor animal and between each pair of animals. Sera were collected from donor and recipient ferrets upon completion of experiments to confirm seroconversion. To ensure no accidental contact or fomite transmission during husbandry procedures, recipient animal sections of the cage were cleaned prior to the donor sides, with three lab personnel participating in each husbandry event to ensure that a clean pair of hands handled bedding and food changes. One cage was done at a time, with a 10 min wait time between cages to remove contaminated air prior to moving on to the next cage. Fresh scrapers, gloves, and sleeve covers were used for each subsequent cage cleaning. Clinical symptoms such as weight loss and temperature were recorded during each nasal wash procedure and other symptoms such as sneezing, coughing, lethargy or nasal discharge were noted during any handling events. Once animals reached 10% weight loss, their feed was supplemented with A/D diet twice a day to entice eating. A summary of clinical symptoms are provided in Table 1.

### 2.7. Serology Assays

Analysis of neutralizing antibodies from ferret sera was performed as previously described [27]. Briefly, the microneutralization assay was performed using 10^3.3^ TCID_50_ of either H3N2 virus incubated with 2-fold serial dilutions of heat-inactivated ferret serum. The neutralizing titer was defined as the reciprocal of the highest dilution of serum required to completely neutralize the infectivity of 10^3.3^ TCID_50_ of virus on MDCK cells. The concentration of antibody required to neutralize 100 TCID_50_ of virus was calculated based on the neutralizing titer dilution divided by the initial dilution factor, multiplied by the antibody concentration.

For hemagglutinin inhibition (HAI) assays, serum samples were pretreated with receptor-destroying enzyme (Seiken) followed by hemadsorption. Briefly, sera were serially diluted 2-fold in a V-bottom plate and mixed with four agglutinating units of virus for 15 min. An equal volume of 0.5% turkey erythrocytes was gently added to each well and incubated for 30 min at room temperature. Agglutination was read and HAI titers were expressed as the inverse of the highest dilution that inhibited four agglutinating units of virus. 

### 2.8. Tissue Sample Collection

Euthanized ferrets were dissected aseptically. Collection of respiratory tissues was performed in the following order: entire right middle lung, left cranial lung (a portion equivalent to the right middle lung lobe), one inch of trachea cut lengthwise, entire soft palate, and nasal turbinates. Tissues were harvested as described in Reference [28] and frozen at −80 °C. Tissue samples were weighed and Leibovitz’s L-15 medium (Invitrogen) was added to make a 10% *w*/*v* homogenate. Tissues were dissociated using an OMNI GLH (OMNI International, Kennesaw, GA, USA) homogenizer and cell debris was removed by centrifugation at 1500 rpm for 10 min. Infectious virus was quantified by TICD_50_ using the endpoint method [25].

### 2.9. In Vitro HA pH Inactivation Assay

Virus stocks were incubated in PBS adjusted to the indicated pH values for 1 h at 37 °C. The infectious titer of the virus after incubation was determined on MDCK cells using the TCID_50_ endpoint titration method [25]. The pH that reduced the titer by 50% (EC_50_) was calculated by regression analysis of the dose–response curves. Each experiment was performed in triplicate in at least two independent biological replicates.

## 3. Results

### 3.1. Viruses with Cell-Culture-Adaptive HA Mutations Replicated Better in MDCK Cells

A previous study by members of our team serially passaged a 6:2 reassortant virus of H3N2 virus with HA and NA genes from A/Perth/16/2009 and internal WSN segments six times in MDCK-SIAT1 cells expressing TMPRSS2. Two mutations in the HA segment, G78D and T212I (H3 numbering), arose after serial passages [23]. To compare the replication kinetics of viruses carrying these cell-culture-adapted mutations, reverse genetics was used to generate both wild-type (WT) and HA mutant virus strains with A/Perth/16/2009 internal gene segments (Figure 1A). The genes for all of these viruses were cloned from a stock of A/Perth/16/2009 (H3N2) virus; however, the HA and NA genes differed between the WT and mutant viruses. Recombinant A/Perth/16/2009 WT (rPerth WT) was generated from reverse-genetic plasmids with native 5′ and 3′ untranslated regions (UTRs) for the HA and NA segments (Figure 1A and Appendix A). However, the mutant strain (rPerth mutant) was generated from plasmids in which the HA and NA were cloned into reverse-genetic plasmids with the noncoding regions from the lab-adapted H3N2 reassortant strain X-31, with the HA also containing the two lab-adaptation mutations G78D and T212I (Figure 1A and Appendix A).

Replication kinetics of rPerth WT and rPerth mutant viruses were examined in MDCK and A549 cells. Each cell line was infected with rPerth WT or rPerth mutant virus at a multiplicity of infection (MOI) of 1.0 or 0.01, and the viral titers in infected cell supernatants were collected at the indicated time points. At the higher MOI in MDCK cells, the rPerth mutant had titers that were 10× greater at 16, 24, and 48 h post-infection (hpi) as compared to those of the rPerth WT (Figure 1B). Similarly, the rPerth mutant grew better than rPerth WT at 24 and 48 hpi under a low MOI (Figure 1C). However, in A549 cells, rPerth mutant replicated better than rPerth WT at 48 hpi only at an MOI of 1.0, and to similar titers under all other conditions (Figure 1D,E). 

Primary HBE cell cultures differentiated at an air–liquid interface mimic the lumen of the human airway because they produce mucus and are permissive to influenza viruses [24,30]. The replication capacity of the rPerth mutant was characterized in three different patient-derived HBE cultures. HBE cells were infected with a low MOI of either rPerth WT or the mutant strain, and released virions were collected at the indicated times post-infection. Regardless of the patient culture, no significant difference in replication between rPerth WT and mutant strains was observed over time (Figure 2). Taken together, these data demonstrate that serial passage of the rPerth mutant on MDCK cells increased replication capacity in MDCK cells, but not in other cell lines or primary airway cultures.

### 3.2. The pH of Inactivation rPerth Mutant Was Higher Than That of WT

Cell-culture adaptation has been associated with a wider range of pH at which HA fuses with the host endosomal membrane, as determined by syncytia assay, compared to WT viruses [31,32,33]. Incubation of virions with acidic PBS at different pH can cause a conformational change in the HA, which results in premature activation of HA and an irreversible loss of viral infectivity [18]. The pH of inactivation for rPerth WT and mutant was determined from the infectious titers of virus solutions incubated in pH-adjusted PBS for 1 h (Figure 3A). The pH of inactivation acts as a surrogate for the pH of fusion [34,35] and is expressed as the EC_50_ from the acid stability assays, which is the pH that reduces the viral titer by 50%. The EC_50_ pH of inactivation for rPerth WT was calculated for each curve and resulted in a pH of inactivation of 5.57, while that of the rPerth mutant was 5.83 (Figure 3A, black dashed lines). This experiment suggests that MDCK cell-adaptive mutations resulted in a virus with decreased acid stability. 

To examine the acid stability of other seasonal human- or swine-origin H3N2 viruses, we evaluated the pH of inactivation for a range of contemporary seasonal and swine H3N2 viruses. As expected, the biological A/Perth/16/2009 had the same pH of inactivation (EC_50_ = 5.57) as the rPerth WT, while the seasonal human viruses tested ranged in pH of inactivation from 5.5 to 4.9 (Figure 3B). These data are consistent with human seasonal H1N1 viruses, which have been shown to have a pH of inactivation < 5.5 [36]. The two H3N2 swine viruses tested had EC_50_ values that were higher (EC_50_ = 5.88 and 5.68) and more similar to the rPerth mutant (Figure 3B). 

### 3.3. Replication of the rPerth Mutant in Ferrets Was Similar to WT

Based on the replication fitness in the various cell-culture systems and differential pH of fusion value for these two viruses, we chose to compare the replication capacity in vivo. Ferrets are an established model for the study of influenza pathogenicity and transmissibility because their lung physiology, receptor expression patterns, clinical signs, and transmissibility patterns resemble those of humans [37]. To determine the infectivity of the rPerth mutant in ferrets, three animals were intranasally inoculated with either rPerth WT or rPerth mutant virus. On day 3 post-infection, the ferrets were sacrificed and respiratory tract tissues (nasal turbinates, soft palate, trachea, right middle lung and left cranial lung) were collected. The amount of viral RNA present in the tissue homogenates was assessed by quantitative PCR and infectious virus. Based on viral RNA quantification, no differences were observed between the recovery of rPerth WT and mutant viruses in any of respiratory tract tissues collected (Figure 4A). Infectious virus was determined on MDCK cells and was primarily observed in the upper respiratory tract (nasal wash, nasal turbinate, and soft palate) (Figure 4B). No significant difference was observed between infectious virus titers from rPerth-WT- and rPerth-mutant-infected ferrets. The discrepancy in detection of viral RNA and infectious virus may have been due to issues with freeze–thaw behavior of the homogenates and titration on MDCK cells rather than MDCK-SIAT cells. Taken together, these observations suggest that cell-adapted H3N2 viruses replicate similarly in ferrets, in line with replication data from HBE cells rather than MDCK cells.

### 3.4. H3N2 Mutant Virus with a Higher pH of Fusion Had Reduced Airborne Transmission to Naïve Ferrets

HA stability is an important factor that impacts transmission efficiency [12,20,21]. To determine whether the pH stability of H3N2 confers an airborne transmission disadvantage and assess the impact of prolonged MDCK cell passage on the transmission efficiency of the rPerth mutant, an airborne transmission study was performed for each virus. Three donor ferrets were infected intranasally with either rPerth WT or rPerth mutant and 24 h later a naïve recipient ferret was placed in the adjacent cage, with directional air flow from the donor to recipient [27,29]. The naïve recipient was exposed for 14 days. Viral titers in nasal washes were collected every other day and seroconversion was determined on day 14 post-infection. rPerth WT transmitted to three out of three recipients, while the transmission efficiency of the rPerth mutant was reduced to one out of three recipients (Table 1). All donors and recipients that shed virus in their nasal washes also seroconverted (Table 1). These results indicate that the cell-adapted rPerth mutant virus has reduced airborne transmission compared to WT.

### 3.5. NA-Segment Noncoding Regions Influence the Virus pH of Fusion

To determine the contribution of the X-31 UTRs to the pH of inactivation, H3N2 viruses with different combinations of HA and NA (WT versus X-31 UTRs) were generated using reverse genetics. Retention of the rPerth HA X-31 UTRs with the head mutations (G78D and T212I) along with the WT NA segment produced a pH of inactivation (EC_50_ = 5.56) which was similar to that of rPerth WT (EC_50_ = 5.57) (Figure 5). However, a rPerth virus strain containing the WT HA segment and the X-31 UTR NA segment had a lower pH of inactivation (EC_50_ = 5.39) compared to rPerth WT (Figure 5), indicating that X-31 UTR flanked NA segment was necessary but not sufficient to alter the pH of inactivation. These results show that both HA (G78D and T212I) and NA X-31 UTR segments are necessary, but are not sufficient alone to increase the pH of inactivation for the rPerth virus.

## 4. Discussion

Cross-species host adaptation is common in influenza viral infections and is the result of selective pressures at the site of replication, cell receptor availability, and compatibility with host transcription and translational machinery. Only H1N1, H2N2, and H3N2 influenza virus subtypes have become established within the human population and transmit person-to-person, while avian H5 and H7 influenza viruses will sporadically infect humans without efficient transmission between individuals. Gain-of-function studies with H5N1 influenza viruses have indicated that mutation of HA to confer α2,6-linkage sialic acid binding, loss of glycosylation within the HA head domain, and increasing HA stability improves airborne transmissibility in the ferret model [20,21]. Evolution of HA is an important player in interspecies transmission and host range expansion. Mutations that stabilize the H5 HA can enhance replication in the upper respiratory tracts of mice and ferrets [34,35,38], but result in a concomitant decrease in replication, virulence, and transmissibility in the natural avian host [39,40]. Conversely, adaptation of human influenza viruses to a murine host required adaptive changes in the HA protein to alter receptor preference and resulted in decreased acid stability from 5.2–5.3 to 5.6–5.8 [41,42,43].

In this study, we demonstrated that MDCK cell-culture adaptation of an H3N2 influenza virus resulted in a virus with lower acid stability than the WT virus. The HA pH of fusion is a known determinant of host adaptation. Consistent with this, we found that human seasonal viruses have an HA pH of fusion of 4.9–5.5, while swine H3N2 viruses have a pH of fusion greater than 5.5. Previous work has shown a similar phenotype with human and gamma clade swine H1N1 viruses [36]. Additionally, gamma clade swine H1 viruses replicate to higher titers in MDCK cells [36], which is consistent with our observations of the rPerth mutant. Interestingly, the observation of increased replication capacity was not recapitulated in HBE cells and ferrets, suggesting that MDCK replication capacity is not a strong correlate to results in relevant organoid cultures or in vivo.

HA is an important determinant of influenza virus transmissibility, as it needs to remain stable as it travels in the environment between hosts [12,44]. HA proteins that are stable at acidic pH have an advantage because they are less prone to being inactivated in their environment [45,46,47]. However, striking a fine pH balance is important for an influenza virus, since a replicative advantage is conferred within the host cell to viruses that encode a less acid-stable HA, as this facilitates efficient viral uncoating in endosomes [48]. However, avian virus HA proteins with a higher activation pH value have some advantages over their more acid-stable human-adapted HA counterparts. During macrophage infection, less acid-stable HA proteins release avian viruses earlier from the endosome to escape lysosomal degradation and allow continued replication [49]. Furthermore, a higher membrane fusion pH has been shown to help avian virus HA proteins avoid detection by the interferon-inducible transmembrane proteins IFITM2 and IFITM3, which restrict virus fusion [50]. Mutations in the head region of the rPerth mutant, which arose during repeated passage in MDCK cells, led to destabilization of HA and raised its activation pH. Similarly, MDCK cell-culture adaption of egg-grown H3N2 X-31 virus caused mutations in or near the fusion peptide, which resulted in higher pH of fusion mutants within a few passages [51]. Improving virus growth in cell culture is important for producing high yields of virus for vaccines, and this enhanced growth is often associated with a broader pH range of virus–host fusion. Site-directed mutagenesis indicates that mutations in the HA2 fusion peptide [32,33,52] and the transmembrane domain [31] can stabilize HA to produce a virus that replicates to higher titers in cell culture. Taken together, these data suggest that cell adaptation correlates with decreased acid stability.

HA stability is only one known determinant of transmission; others include receptor binding, polymerase activity, and virus morphology [53]. Specific mutations in the receptor-binding site of HA can cause altered receptor preferences [54]. Although amino acids T212 and G78 lie outside the receptor-binding pocket, further studies are needed to determine their impact on receptor specificity and avidity, which may also contribute to the observed decrease in airborne transmission. Influenza viruses are pleiomorphic structures, with HA and NA packed closely but irregularly distributed on the surface of the virus particle [55,56,57]. NA is typically present in much smaller quantities that HA and it has been estimated that influenza virus particles have roughly 300 HA and 20–40 NA proteins [56,58]. HA and NA have opposite functions and a fine balance is required for efficient virus replication, as HA binds to sialic-acid-containing receptors and NA removes sialic acid from host cells. Mutations that alter the NA enzymatic active site or stalk length have been linked to imbalance of the HA/NA relationship [59,60,61]. The pH of fusion phenotype for the rPerth mutant was mapped to both the HA G78D and T212I changes and X-31 UTRs flanking the NA segment (Figure 1A). This observation suggests an additional aspect of HA/NA balance whereby the noncoding regions of the NA gene segment can influence HA function. The viral gene segment UTRs are essential promoter elements required for initiation of viral replication and transcription. Viral gene expression requires interaction of the RNA-dependent RNA polymerase complex with both conserved sequence elements in the UTRs. Mutations in the UTR promoter region have been shown to modulate gene expression and downstream protein expression [62,63]. Therefore, changes in the NA UTR may alter the amount of NA synthesized during a viral infection and ultimately impact the amount of the protein present on a virion. Decreased incorporation of NA in virions has been observed to negatively impact NA activity and compensate for functional differences in HA [64,65]. The balance between HA and NA is critical for influenza virus fitness and a future avenue of research will be to determine the expression levels of mRNA and protein, and how these potential differences might be affected by different UTRs.

## Figures and Tables

**Figure 1 viruses-13-00719-f001:**
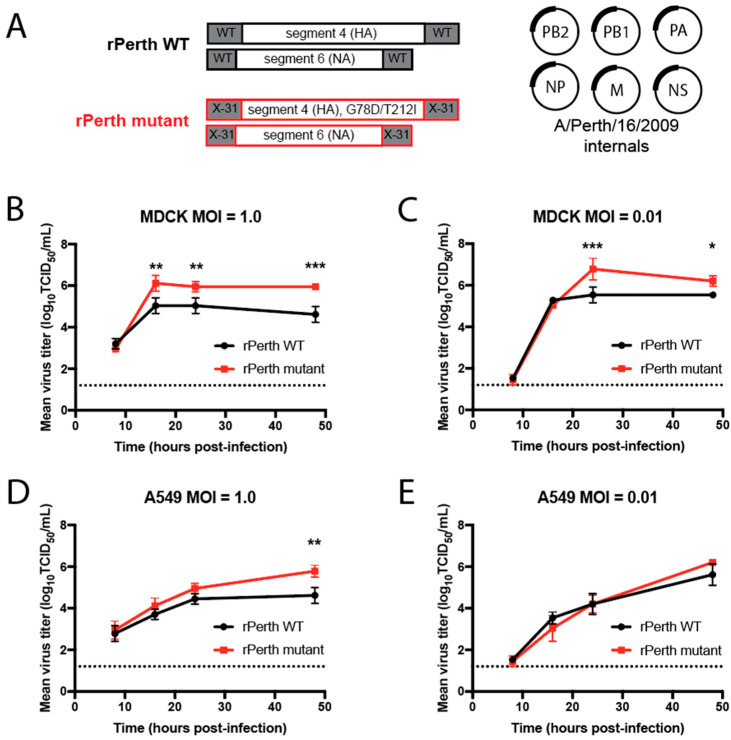
In vitro replication of cell-culture-adapted H3N2 virus. (**A**) Schematic representation of recombinant (r) A/Perth/16/2009 wild-type (WT) and mutant virus constructs. MDCK (**B**,**C**) and A549 (**D**,**E**) cells were infected with rPerth WT (black circles) and HA mutant (red squares) viruses at MOIs of 1.0 or 0.01. Cells were infected in triplicate and supernatants were collected at the indicated times. Virus titers were determined on MDCK cells using TCID_50_ assays. Graphs are representative of three independent experiments. Two-way ANOVA was used to determine statistical significance (* *p* < 0.05, ** *p* < 0.005, *** *p* < 0.0005). The dashed line denotes the limit of detection for the titration assay.

**Figure 2 viruses-13-00719-f002:**
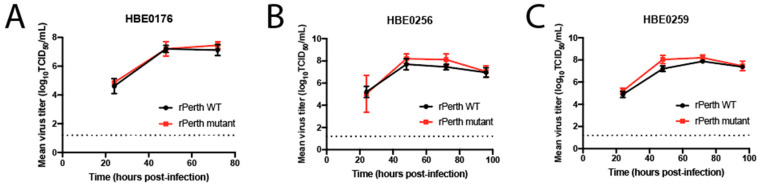
In vitro replication of rPerth WT and HA mutant in human bronchiole epithelial (HBE) cells. Cells from three different patient cell lines, (**A**) HBE0176, (**B**) HBE0256, and (**C**) HBE0259, were infected at 10^3^ TCID_50_ per well with rPerth WT (black circles) or rPerth mutant (red squares). The apical supernatant was collected at the indicated time points and virus titers were determined on MDCK cells using TCID_50_ assays. The experiments were performed in triplicate.

**Figure 3 viruses-13-00719-f003:**
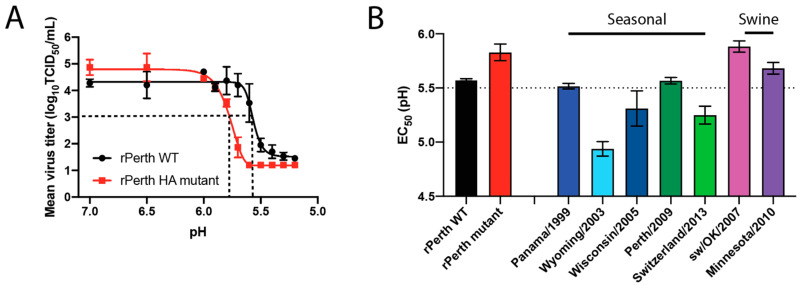
rPerth mutant had decreased HA stability. (**A**) rPerth WT (black line) and mutant (red line) were treated in pH-adjusted PBS for 1 h at 37 °C. Remaining virus titers were determined by TCID_50_ assay. The experiment was performed in triplicate, and representative data are shown. The data were fit with an asymmetric sigmoidal curve to determine the EC_50_. The limit of detection was 1.2 log_10_ TCID_50_/mL. (**B**) Seasonal and swine H3N2 viruses were incubated in PBS of different pHs for 1 h at 37 °C, with experiments performed in triplicate. The remaining virus titers were determined by TCID_50_ assay and the EC_50_ values were plotted using regression analysis of the dose–response curve. The reported mean (±SD) corresponds to two independent biological replicates, each performed in triplicate.

**Figure 4 viruses-13-00719-f004:**
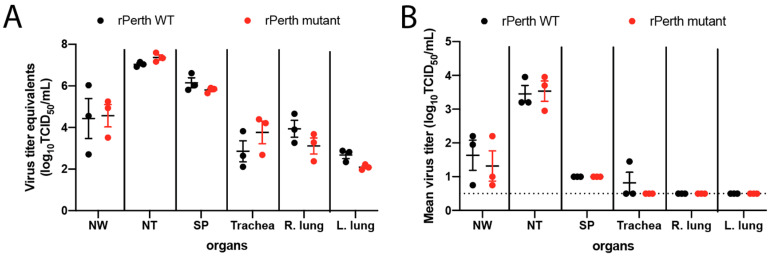
Quantification of H3N2 virus in respiratory tissues of infected ferrets. Ferrets were intranasally infected with 10^6^ TCID_50_ in 0.5 mL of rPerth WT (black circles) or rPerth mutant (red circles) and were sacrificed on day 3 post-infection. NW—nasal wash, NT—nasal turbinate, SP—soft palate. (**A**) RNA was isolated from the indicated respiratory tract organ homogenates and qPCR for a region of the M segment was performed to quantify the relative amounts of influenza virus, normalized to RNA isolated from a virus stock with a known titer. (**B**) Titration of tissue homogenates to quantify infectious viral titers. Each dot represents a single animal and mean line ±SEM is depicted.

**Figure 5 viruses-13-00719-f005:**
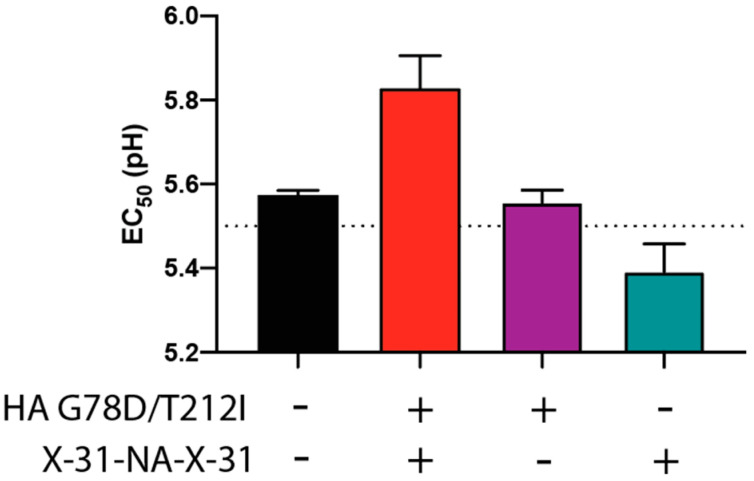
Increased pH of inactivation requires X-31 UTRs in NA segment and HA mutations. The indicated mutant H3N2 viruses were incubated in PBS of different pHs for 1 h at 37 °C, with experiments performed in triplicate. The remaining virus titers were determined by TCID_50_ assay and the EC_50_ values were plotted using regression analysis of the dose–response curve. The reported means (±SD) correspond to two independent experiments performed in three technical replicates.

**Table 1 viruses-13-00719-t001:** Airborne transmission of rPerth WT and mutant viruses.

Virus	Exposure Time	Status	Transmission Efficiency	Temperature Increase *	Weight loss*	H3N2 Microneutralization Titers ^	HAI Titers ^^
rPerth WT	14 days	INF		0/3	0/3	2560, 250, 2560	2560, 250, 2560
		Naïve	3/3	1/3	0/3	1280, 2560, 1280	1280, 2560, 1280
rPerth mutant	14 days	INF		1/3	0/3	2260, 1010, 2260	1280, 1280, 690
		Naïve	1/3	1/3	0/3	<20, <20, 1010	<10, <10, 2560

* Temperature increase is defined as >1.5deg from day 0. Weight loss determined as > 10% of day 0. ^ Antibody titers of day 14 are presented. All day 0 sera had a titer <20. ^^ Antibody titers of day 14 are presented. All day 0 sera had a titer <10.

## Data Availability

Data is contained within the article and supplementary material. All of the plasmid sequences are available as Snapgene files on FigShare at https://figshare.com/projects/Cell_Culture_Adaptation_of_H3N2_Influenza_Virus_Impacts_Acid_Stability_and_Reduces_Airborne_Transmission_in_Ferrets/112005 (accessed on 21 April 2021).

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
