# Peer review of "Cell-Culture Adaptation of H3N2 Influenza Virus Impacts Acid Stability and Reduces Airborne Transmission in Ferret Model"

_viruses, 2021, doi:10.3390/v13050719_

Round 1
Reviewer 1 Report
The authors describe the cell culture adapted H3N2 virus show lower HA stability and reduced transmission efficiency. These results suggest that higher pH of fusion of H3N2 virus cause decreased airborne transmission.
The paper is well described, the results are supported by the data which are clearly presented in figures and tables, however, some important details are unclear.
1. The manuscript does not describe the effect of HA mutation to sialic acid binding ability. As manuscript described, receptor binding ability of HA is also important for viral replication and transmission efficiency. Is there any difference on binding affinty or spesificity to a2,3 and a2,6 sialic acid between WT virus and mutant virus?
2. How UTRs of NA affect to pH stability of viruses? Does change of UTRs of NA affect the mRNA expression level of NA?
Minor comment
3. In Figure 3, why do viruses even treated in pH lower than 5.5 show TCID50 titer? Do the viruses maintain infectivity in such lower pH?
Author Response
We provide a point-by-point response to reviewers 1 comments in the attached document.

Reviewer 2 Report
In the manuscript entitled "Cell culture adaptation of H3N2 influenza virus impacts acid stability and reduces ferret airborne transmission", Valerie et al., described the importance of HA pH stability on cell culture adaptation of H3N2 influenza. This is an important piece of work, fairly well written. However, there are number of confusing and poorly constructed sentences throughout the manuscript. I would suggest authors let someone expert in academic writing proofread the manuscript.
Abstract:
-Avoid the use of too many personal pronouns in Abstract or elsewhere. It is not wrong but looks untidy.
-In addition, we observed that this mutant H3N2 virus replicated to higher titers in cell culture but had reduced airborne transmission in the ferret model. These data demonstrate that, like H1N1 HA, the pH of fusion for H3N2 HA is a determinant of efficient airborne transmission. Surprisingly, we demonstrate that the NA segment noncoding regions can impact the pH of fusion of reassortant viruses.
-I would suggest authors further polish the introduction part; clearly describe the questions, and concerns that you would like to address. Some sections are well written; however, it could be much better.
If Ref. [23] is one of your studies, then revise the following sentences
" A previous study using a 6:2 reassortant virus of H3N2 virus with HA and NA Perth/16/2009 and internal WSN segments identified two mutations in the HA segment, G78D and T212I (H3 numbering), which arose after six serial passages in MDCK-SIAT1 cells expressing TMPRSS2”
To compare the replication kinetics of viruses carrying these cell culture-adapted mutations, reverse genetics was used to generate both wildtype (WT) and mutant virus strains used in this study.
-Our transmission caging setup is a modified Allentown ferret and rabbit bioisolator cage similar to those used in ?
Author Response
We provide a point-by-point response to the reviewers comments- attached.
